# Evaluating Fire Performance: An Experimental Comparison of Dovetail Massive Wooden Board Elements and Cross-Laminated Timber

**Hüseyin Emre Ilgın *** , **Markku Karjalainen** , **Mika Alanen** and **Mikko Malaska**

Faculty of Built Environment, Tampere University, P.O. Box 600, FI-33014 Tampere, Finland;
markku.karjalainen@tuni.fi (M.K.); mika.alanen@tuni.fi (M.A.); mikko.malaska@tuni.fi (M.M.)
* Correspondence: emre.ilgin@tuni.fi

**Abstract:** The use of adhesives and metal connectors is vital in engineered wood product (EWP) composition. However, the utilization of adhesives poses sustainability and recyclability challenges due to the emission of toxic gases. Similarly, metal fasteners negatively impact the disposal, reusability, and recyclability of EWPs. An alternative solution that exclusively employs pure wood, known as dovetail massive wooden board elements (DMWBEs), eliminates the need for adhesives and metal fasteners. This paper presents an experimental comparative assessment of the fire/charring performance of DMWBEs and cross-laminated timber (CLT). Model-scale test specimens measuring 200 mm in thickness, 950 mm in width, and 950 mm in length were vertically tested according to EN 1363-1. The charring behavior of DMWBEs closely resembled that of solid timber, with only a slight increase in the charring rate. Charring primarily occurred in the third lamella layer out of five, with no observable flames or hot gases on the unexposed side. The dovetail detail effectively prevented char fall-off with the tested lamella thickness. CLT specimens exhibited a notable rise in the charring rate due to the fall-off of the first lamellae layer.

**Keywords:** dovetail massive wooden board elements; CLT; fire performance; char depth; charring rate

## 1. Introduction

Thanks to its diverse technical advantages, e.g., stiffness, dimensional stability, and environmentally friendly characteristics, EWPs have progressively gained traction as a building material in the construction industry since the 1990s [1–4]. Their competitiveness has notably increased in the realm of tall building construction [5,6], exemplified by the utilization of EWPs in prominent structures [7,8] like the 49-m-high Treet in Bergen, Norway (refer to Figure 1) and the 48-m-high Lighthouse Joensuu in Joensuu, Finland (refer to Figure 2) [9].

Adhesives and metal connectors are commonly employed in EWPs to replace conventional timber-to-timber joints in modern wooden structures [10]. In this regard, adhesive bonding holds significant importance and serves a crucial role in EWPs [11]. Adhesives contribute to the structural integrity and lightweight nature of the building, while also mitigating issues related to timber preservation and preventing dimensional changes due to fluctuations in humidity levels. However, the use of adhesives poses challenges in terms of sustainability, recyclability, and overall ecological impact, primarily due to the emission of toxic gases such as formaldehyde throughout their lifespan [12]. Furthermore, despite ongoing advancements in this research field, substantial inquiries persist regarding the viability of environmentally friendly bio-based adhesives [13]. Similarly, while metal fasteners are indispensable in EWPs, their utilization negatively impacts end-of-life disposal options, reusability, and recyclability [14].

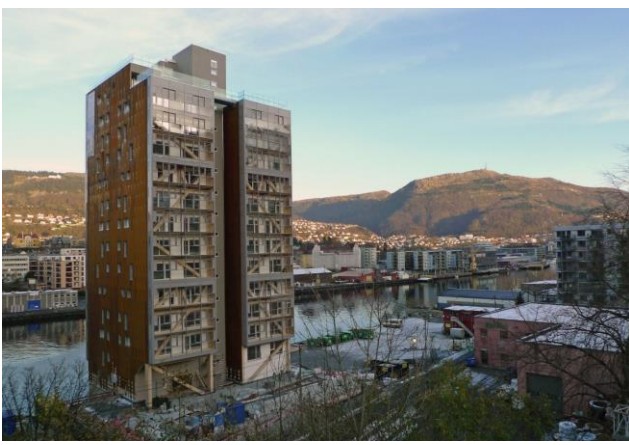

**Figure 1.** Treet (Photo courtesy of ARTEC).

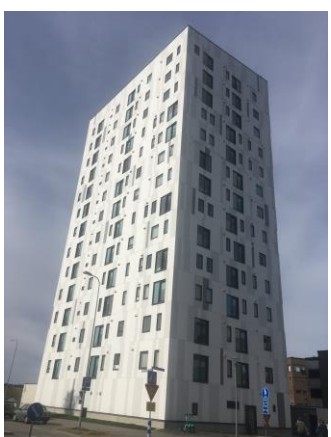

**Figure 2.** Lighthouse Joensuu.

An alternative solution exists in the form of dovetail massive wooden board elements that solely rely on pure wood and eliminate the need for adhesive and metal fasteners (refer to Figure 3). To date, substantial scholarly inquiries have been carried out regarding the technical characteristics of timber materials, encompassing various construction methods employing EWPs, as documented in the existing literature (e.g., [15–18]). However, limited investigation has been dedicated to dovetail massive wooden board elements (DMWBEs), with the available literature predominantly focusing on structural analyses of connection details rather than comprehensive assessments of load-bearing components like floor slabs [19,20]. Consequently, our understanding of the potential of DMWBEs, particularly concerning their environmental impact and recyclability, remains incomplete [21].

This research investigates the fire performance of DMWBEs, composed of interconnected wooden lamellae using one of the most ancient methods of joining. This manufacturing technique presents a solution that is devoid of adhesives and metal connectors, ensuring the absence of any emissions of harmful substances [22]. Since this represents a novel approach, the available data regarding the technical and structural performance are exceedingly scarce, necessitating further investigation, particularly in areas such as dimensional stability [23,24]. Under the umbrella of the DoMWoB project (Dovetailed Massive Wood Board Elements for Multi-Story Buildings) (refer to the Acknowledgment section), a comprehensive plan was formulated to conduct technical performance evaluations encompassing structural (bending) performance, fire performance, air permeance, and sound insulation tests. The primary objective behind these tests was to advance the development of DMWBEs as a globally viable substitute for conventional EWPs.

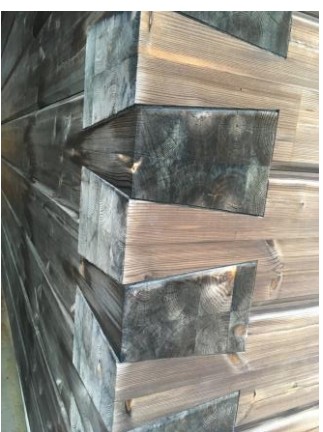

**Figure 3.** Dovetail connection detail in a wall.

While there exist various models for evaluating the fire/charring performance of CLT assemblies and other laminated materials like bamboo in Europe and North America (e.g., [25–27]), researchers are exploring different approaches to enhance their fire performance due to the growing market demand as detailed below. Among the notable investigations, Yasir et al. [28] conducted empirical fire tests on four vertically loaded Cross-Laminated Timber (CLT) wall panels composed of Irish spruce. The outcomes reveal that employing protective claddings using Fireline gypsum plasterboard and a composite of plywood and Fireline gypsum plasterboard resulted in a significant prolongation of the charring process in CLT panels, extending by approximately 30 and 44 min, respectively. The research conducted by Hopkin et al. [29] contributes to an enhanced comprehension of the fire behavior of exposed CLT within expansive enclosures. A configuration resembling an office setup was simulated using a sizeable enclosure measuring 3.75 by 7.6 by 2.4 m in height, constructed with noncombustible blockwork walls and featuring a sizable opening along one of its lengthy sides. The findings underscore the significance of evaluating the influence of ceiling protrusions, such as down-stand beams, as evidenced by discernible distinctions in radiative heat flux affecting both the ceiling and floor in the contrasting scenarios. This insight becomes particularly pertinent in the context of large contemporary open-plan office enclosures, potentially leading to variations in the pace of fire spread within an enclosure and the time required for auto-extinguishment of flaming combustion. Jin et al. [30] delved into the role of combustible CLT in contributing to the fire load within a CLT compartment, as well as the dynamic effects of CLT lay-up under compartment fire conditions. To investigate this, four distinct CLT compartment fire tests were conducted, varying the fire areas on the inner surface and the arrangement of CLT layers. The proportions of the directly exposed CLT surface to fire were set at 0%, 19.8%, 36.4%, and 87.6% of the inner compartment surface. The experimental outcomes underscore the notable influence of the CLT fire area on the evolution of the compartment fire and its heat release characteristics. The combustibility of the CLT material was observed to significantly amplify both the burning rate and the rate of heat release from the fire. Moreover, the presence or absence of detached charred layers exhibited a pronounced impact on the progression of the fire. Furthermore, a degree of randomness was identified in terms of both the timing and the extent of the detachment of charred layers. In Lv's study [31], the focus was on gaining more comprehensive insight into the charring characteristics of Cross-Laminated Bamboo (CLB). To achieve this, three distinct sets of CLB slabs were subjected to diverse fire-resistant treatments: no treatment, application of fire-retardant coatings through painting, and impregnation with flame retardants. The slabs were then subjected to fire exposure on a single side, with furnace temperatures adhering to the specifications outlined in ISO 834-1. The outcomes of the tests revealed that, when exposed to identical fire durations, the extent of charring in the CLB slabs displayed a declining trend based on the applied fire-resistant methods. Specifically, the sequence of decreasing charring

degree among the different treatments was as follows: absence of fire-resistant treatment > impregnation with a monobasic ammonium phosphate solution > application of fire-retardant coatings through painting. Kontis et al. [32] conducted an extensive evaluation of contemporary experimental investigations that sought to elucidate the fire characteristics of CLT components. This comprehensive analysis encompassed a substantial volume of test outcomes derived from diverse experimental setups, encompassing varying scales such as cone calorimeter trials (50 tests), standard fire resistance furnace assessments (90 tests), and fire compartment experiments (20 tests). The study involved a comparative scrutiny of these results. The principal focus of the study was to examine the influence of key material and design parameters on critical fire performance metrics. By meticulously analyzing the compiled experimental findings, certain overarching trends that are consistently observed across a majority of instances were discerned.

It is important to highlight that, in the existing literature, there is an absence of research concerning the fire/charring performance of nonadhesive and nonmetallic wood products. Furthermore, the available studies on fire performance are extremely limited, and they solely pertain to dovetail wooden elements at the joint detail level (e.g., [33]).

When wood is exposed to fire, an initial heating phase is followed by pyrolysis, a thermal degradation process that typically begins around temperatures of 260–300 °C [34]. Pyrolysis leads to the generation of combustible gases and the loss of timber mass due to moisture evaporation and migration [35]. As pyrolysis continues, a char layer forms on the surface of the wood, gradually thickening. This char layer acts as a natural insulator, providing the underlying wood with low effective thermal conductivity [36]. The degree of charring is commonly quantified by the charring rate (β), which indicates the depth of charring within a specific time frame. Thick wood elements exhibit a predictable slow charring rate, contributing to favorable fire ratings for wood members [37]. Despite wood being a combustible material and experiencing a reduction in cross-sectional size during a fire, the noncombustible inner part, known as the inner core, retains its mechanical strength and exhibits resistance similar to its pre-fire condition. This characteristic enables heavy timber systems to maintain significant structural strength over extended periods during a fire [38]. In the case of CLT, the charring rate can often be higher than that of solid wood due to defective bonds and gaps between layers, which accelerate the charring process. However, studies have shown that CLT behaves similarly to timber in terms of charring rates, as long as delamination does not occur [39]. Delamination refers to the separation of individual charred layers, which can significantly increase the charring rate and introduce additional fuel, potentially leading to a second flashover event [40,41].

A continuous research endeavor underway at Tampere University is currently dedicated to examining the behavior of DMWBE structures under typical temperature circumstances and when subjected to standard fire exposure. In this manuscript, an experimental comparative analysis on fire performance was performed, utilizing CLT as a reference, constituting a significant phase in the evaluation of DMWBE technical capabilities within the framework of the DoMWoB project. Test specimens of the model scale, possessing dimensions of 200 mm in thickness, 950 mm in width, and 950 mm in length for both CLT and DMWBEs (see Figure 4), were subjected to vertical testing in accordance with EN 1363-1 [42].

The fundamental rate of char progression in a pine or spruce solid timber element is commonly established at 0.65 mm/min [43]. In the case of CLT structures, the charring rate values are contingent upon numerous factors. Typically, these values exceed those of solid wood due to the inherent inability of all utilized adhesives to impede heat-induced delamination and premature detachment of the char layer as the char depth surpasses the bond lines between laminations. Conversely, within the DMWBE structure, the absence of adhesives and the implementation of a dovetail detail effectively mitigate the occurrence of untimely char detachment.

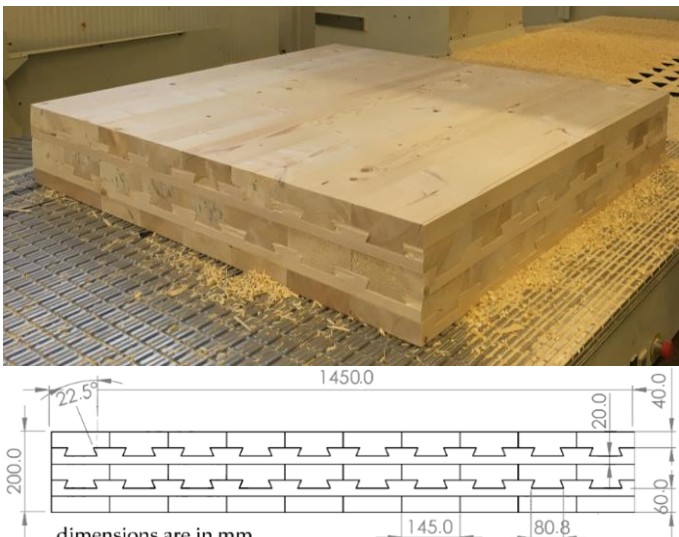

**Figure 4.** The prototype served as the specimen for conducting charring performance tests.

This study encompassed two fire experiments carried out on specimens of DMWBE and CLT with comparable lamella thicknesses. The primary objective was to examine the efficacy of the dovetail detail in averting delamination within the DMWBE structure under fire conditions, as well as assessing the structural integrity of DMWBEs and their ability to impede the transmission of flames and hot gases throughout the testing process. Additionally, estimates of the charring rate were derived based on temperature measurements obtained within the specimens. The findings and observations were juxtaposed with those of CLT specimens fabricated using a polyurethane adhesive. The experimental tests and the principal conclusions are outlined in this study.

## 2. Materials and Methods

This part outlines the fire examination conducted on panels of DMWBEs and CLT at Tampere University's Fire Laboratory, with a focus on evaluating the char depth and charring rate.

### 2.1. Test Specimens

In this experimental study, the primary objective was to investigate and compare the charring development of DMWBE and CLT panels. These tests were conducted without applying any external load. Test methodology and experimental arrangement similar to previous research were followed [34,44,45]. The tests were conducted on a 200 mm × 950 mm × 950 mm panels. Since the temperatures were measured with thermocouples inserted from the side of the specimen, a larger size was considered unnecessary.

#### 2.1.1. Dovetail Test Specimens

DMWBEs were fabricated at Lapland Vocational College, situated in Finland [46]. A 5-axis CNC machine (refer to Figure 5) equipped with the NUM operating system and compatible SOLIDWORKS computer application was utilized to manufacture the two test specimens. The CNC post-processor methodology was implemented, establishing an integrated environment that facilitated the various stages of finishing, toolpath optimization, and G-code simulation for the manufacturing process [47]. The moisture content of the dovetail boards during fabrication ranged from 10% to 12%. However, the production process encountered challenges primarily related to the absence of a mass production line, resulting in a relatively lengthy production time, as well as the requirement for different types of tools such as blades, and the necessity to remove dust during manufacturing.

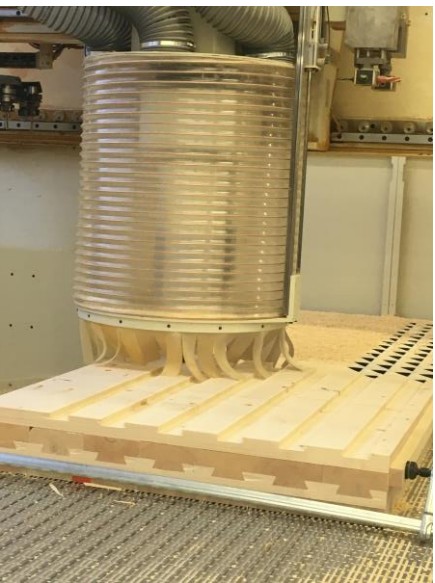

**Figure 5.** The manufacturing of DMWBEs at Vocational College Lapland in Kemi, Finland involved the utilization of a CNC machine equipped with a 5-axis configuration.

To investigate the fire-resistant characteristics of DMWBEs, two separate panels were manufactured. Each panel was initially constructed with measurements of 200 mm in thickness, 1015 mm in width, and 1450 mm in length (see Figure 6). Subsequently, they were trimmed to their final dimensions of 200 mm in thickness, 950 mm in width, and 950 mm in length. The panels were constructed using Norway Spruce wood of C24 PS strength class. The moisture content of the specimens subjected to testing was determined to be 10.3% at the time of the experiment. The dry density of specimens was 362 kg/m$^3$.

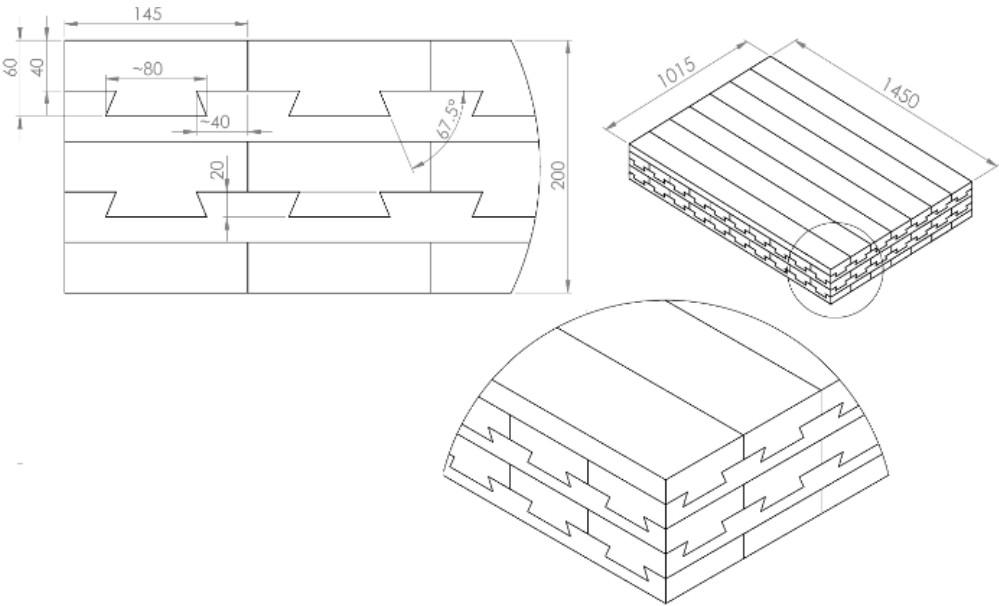

**Figure 6.** DMWBE test sample production drawings.

### 2.1.2. CLT Test Specimens

CLT panels intended for comparison with DMWBE panels were fabricated at CLT Plant Oy located in Finland [48]. To evaluate the fire resilience of CLT, two separate panels underwent experimental analysis. Each panel possessed measurements of 200 mm in thickness, 950 mm in width, and 950 mm in length (refer to Figure 7), which mirrored the

specifications of the DMWBE panels. The individual lamella dimensions were measured to be 145 mm by 40 mm. For the construction of the CLT panels, M1 class polyurethane adhesive obtained from Kiilto Oy in Tampere, Finland was utilized. The adhesive was applied to all four surfaces of each lamella. The CLT panels utilized boards comprised of Norway Spruce timber classified under the C24 PS strength category. The moisture content of the specimens subjected to testing was determined to be 9.6% at the time of the experiment. The dry density of specimens was 428 kg/m$^3$.

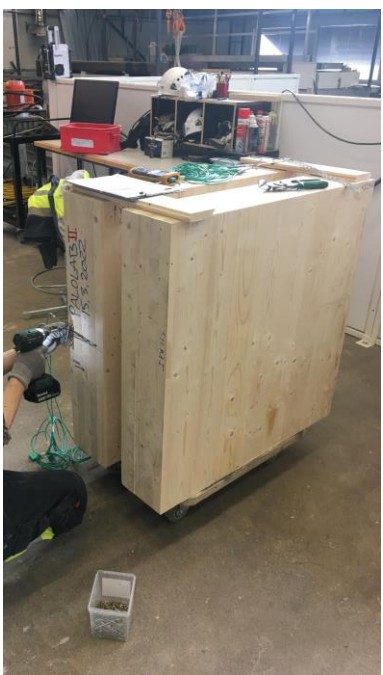

**Figure 7.** CLT specimens were prepared for the fire performance test conducted at the Fire Laboratory of Tampere University located in Tampere, Finland.

### 2.2. Test Set-Up

For each trial, two specimens with comparable structures were affixed in a vertical position to a supporting framework constructed from aerated concrete blocks, as depicted in Figure 8. The specimens were oriented with their outer lamella layers arranged vertically.

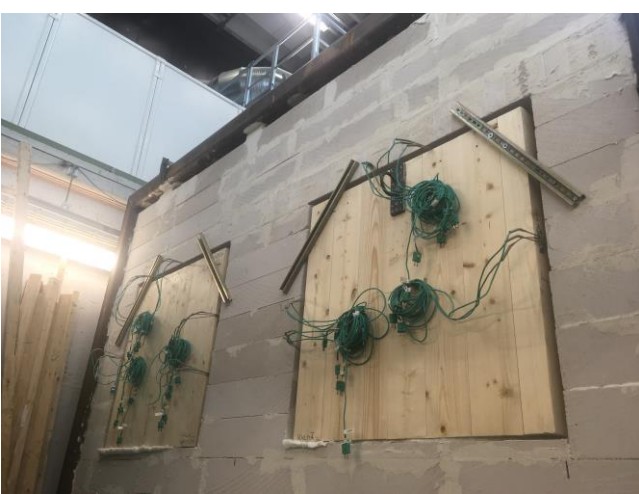

**Figure 8.** The specimens were affixed to a supporting structure composed of aerated concrete blocks. (Unexposed surface).

The experiments were carried out following the guidelines outlined in EN 1363-1 (2020). Throughout the testing process, parameters such as furnace temperature, specimen temperatures, oxygen levels within the furnace, and pressure differentials between the furnace and test area were meticulously monitored. The pressure at the top edge of the specimen was maintained at a fixed value of 20 Pa. To determine the oxygen concentration within the central region of the furnace chamber, a Dräger EM200-E multi-gas detector was employed.

The evaluation of char depth and charring rate relied on temperature measurements conducted within the specimens throughout the test. Given that the primary objective of these experiments was to observe the fire performance of the novel adhesive-free construction method and determine whether the dovetail detail could impede panel delamination, it was deemed adequate to focus on temperature measurements solely at the primary interfaces between the lamella layers. In each specimen, the temperatures were measured in three interfaces between lamella layers and at five different locations as shown in Figure 9. The charring rate was assessed at 10 different positions within CLT element and another 10 positions within the DMWBE element. This analysis involves a comparison of products, and it was determined that conducting 10 measurements for each product provided adequate data. In prior research [33,43,44], analyses were conducted using either three or nine measurement locations within each product. Figure 9 illustrates schematic diagrams depicting the arrangement of thermocouples in both DMWBEs and CLT elements. By combining the results of the three overlapping thermocouples in each of the five location, a graph showing the development of the char depth can be produced. The charring rate can be determined from the graph.

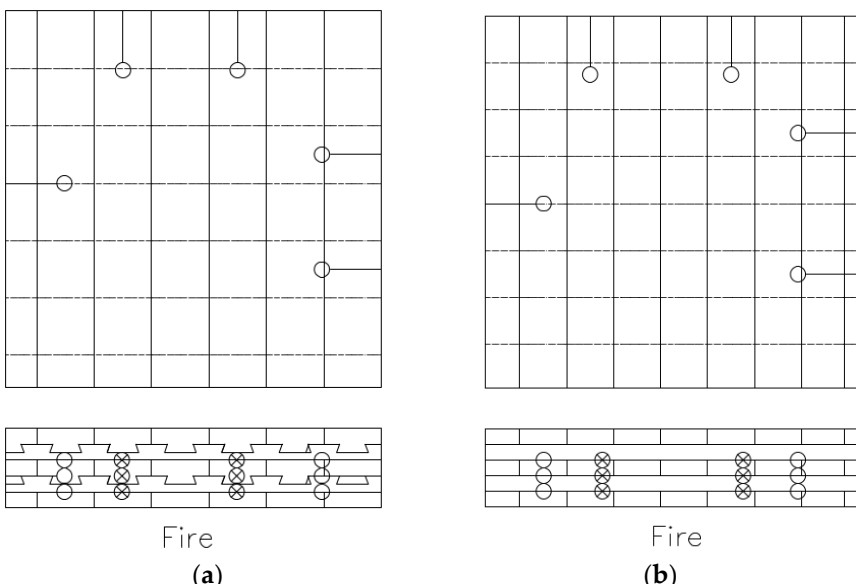

**Figure 9.** The vertical view and cross-sectional representation of the CLT specimens, along with the locations of the embedded thermocouples, are illustrated as follows: (**a**) DMWBE panels and (**b**) CLT panels.

Shielded 3 mm diameter Type-K thermocouples were installed into 3.5 mm diameter holes drilled from the side face of the specimen. The holes were drilled to a depth of 150 mm along the interface between the layers.

## 3. Results

Both tests were terminated and the burners were shut off 142 min after the commencement of the test, and the specimens were immediately extinguished. The furnace temperature adhered to the prescribed temperature–time curve [21]. In both trials, the oxygen concentration within the furnace chamber hovered around 5% for the initial 60 min,

as depicted in Figure 10. This coincided with the charring of the first lamella layer. For the DMWBE, the oxygen content steadily declined thereafter, reaching approximately 2.5% at the 120 min mark. Conversely, for CLT, the oxygen content rapidly plummeted to zero at the 65 min mark, remaining at this low level for 15 min. Simultaneously, extensive sections of the first lamella layer were observed to detach through visual observations from the furnace camera. Subsequently, the oxygen content swiftly rose again to 4% and remained at this level until 117 min, after which it decreased once more to zero. This pattern aligns well with the visual observations of the detachment of the second lamella layer.

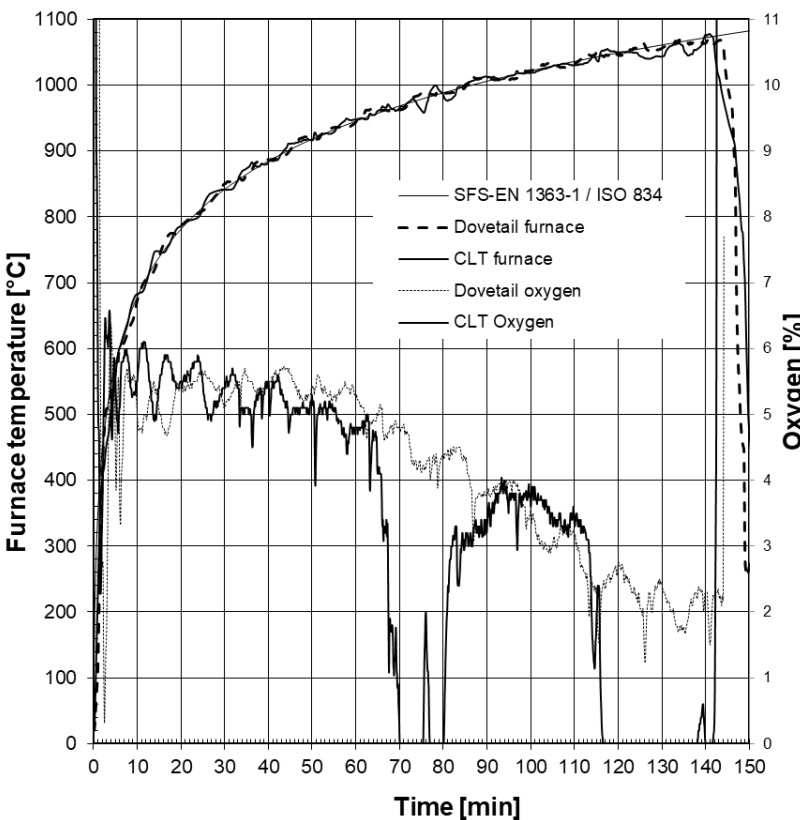

**Figure 10.** Furnace temperatures and oxygen concentrations were measured during the fire tests.

The findings indicated that both panel products effectively obstructed the transfer of flames and hot gases across the structures. In each specimen, the char front was observed to reside within the third lamella layer by the conclusion of the 140 min test duration.

The determination of char depths in the panel product relied on temperature measurements conducted within the specimen during the test, with a reference charring temperature of 300 °C for the wood. The temperature developments at different depths of the DMWBE and CLT panel cross-sections are shown in Figure 11. A curve represents the temperature development at a thermocouple. Ten thermocouples were used at each depth considered. Based on Figure 11a, the DMWBE panel temperatures at 40 mm exceeded the 300 °C limit in the first and in the last thermocouple at 59 and 71 min, respectively. Figure 11b shows that the first 40 mm thick lamella of the CLT panel behaved very similarly. After this, the panels behaved differently and the temperatures at a depth of 80 mm rose significantly faster in the CLT panel. This was caused by the failure of the bonding adhesive layer, resulting in the falling off the char layer protecting the remaining uncharred layers behind it. Based on the temperature measurements, the dovetail detail was able to limit the premature char fall off.

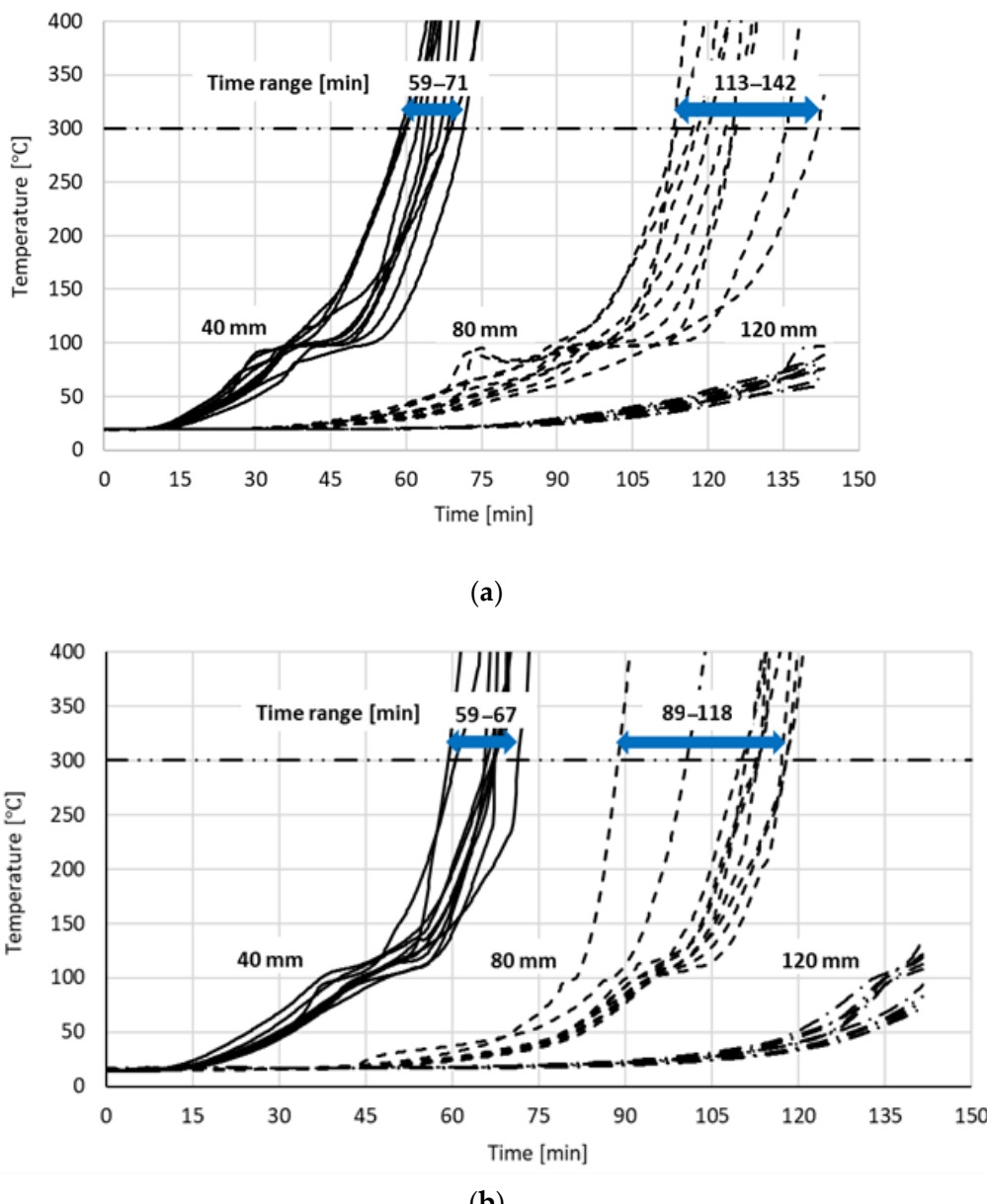

**Figure 11.** Temperature development in (**a**) DMWBE and (**b**) CLT panel cross-sections at given depths.

Figure 12 shows the charring depth as a function of time for DMWBE and CLT panel specimens. Grey lines represent the char depth development based on temperature measurements in one set of three overlapping thermocouples and the black line the mean char depth based on the ten sets. The mean char depth is determined up to the point where the depth in the first thermocouple set exceeds 80 mm. The tests were terminated before the char front reached the level of the last thermocouples at the depth of 120 mm. Therefore, the mean char depth can no longer be determined after the char front in the first thermocouple set exceeds 80 mm. From the figure, it can be seen that the charring depth development of the DMWBE panels is linear and that the variation of the results (grey lines) is smaller than in the CLT structure. Figure 12 also shows that there is a clear change in the charring rate of the CLT structure when the char front progresses to the second lamella layer.

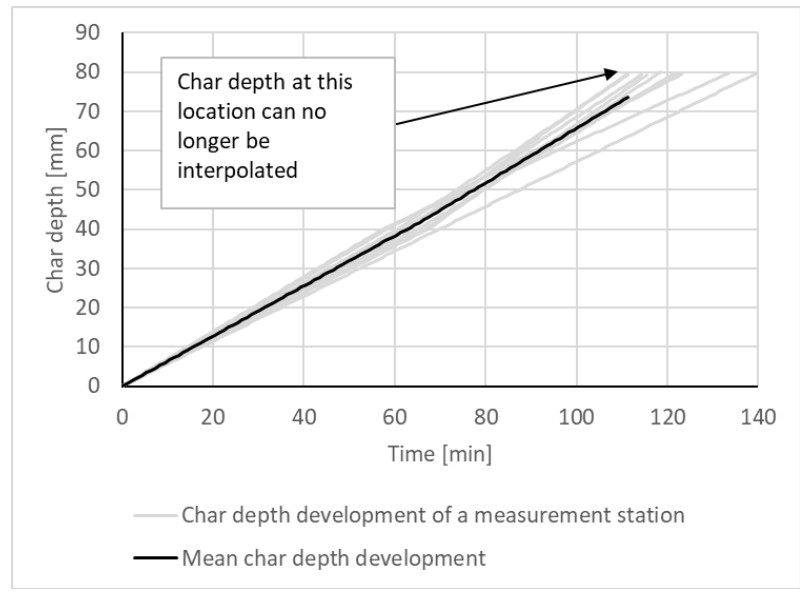

**(a)**

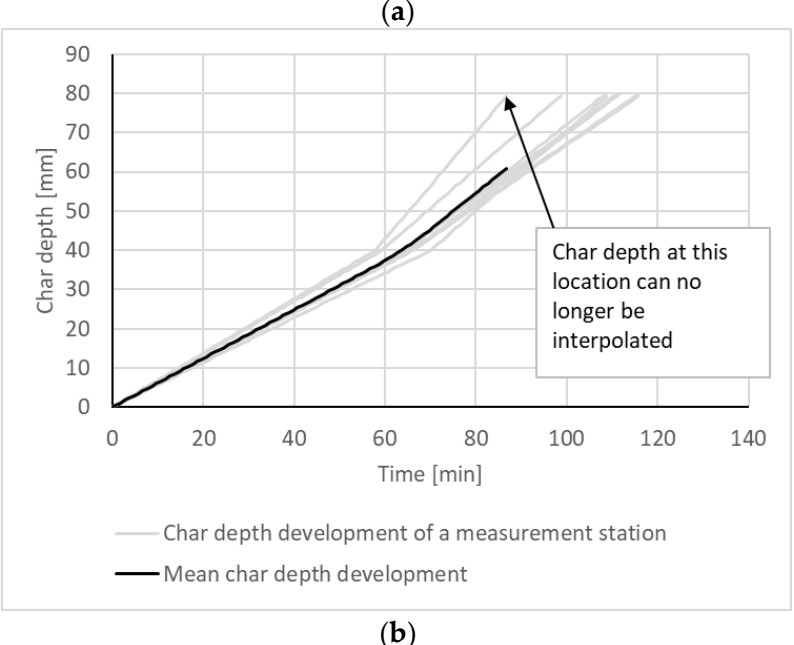

**(b)**

**Figure 12.** Charring depth as a function of time for (**a**) DMWBE and (**b**) CLT panels. Grey lines represent the char depth development based on temperature measurements in one set of three overlapping thermocouples and the black line the mean depth based on the ten sets.

## 4. Discussion

Figure 13 compares the average charring depths observed for both the DMWBE and CLT panels. Additionally, Figure 13 illustrates the advancement of charring depth in accordance with the specified charring rate of 0.65 mm/min for solid wood. The findings indicated that the charring performance of the DMWBE panels closely aligns with that of solid wood, while the charring rate of the CLT panels begins to escalate following the detachment of the first lamella layer at 60 min. As the tests were concluded prior to the char front extending beyond 120 mm at any thermocouple position, the average charring depths can only be determined up to the point where the temperature of a thermocouple at 80 mm initially surpasses 300 °C. This explains why the curve representing the average charring depth for the CLT panel ceases at 86 min.

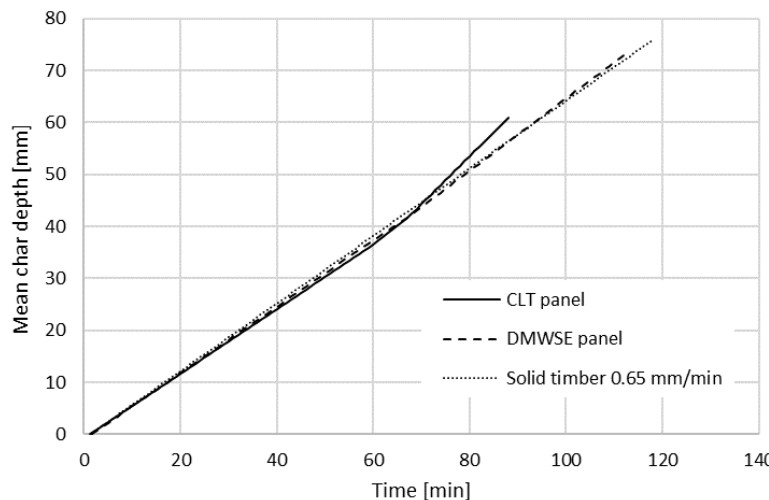

**Figure 13.** Mean charring depths for DMWBE and CLT panels. For comparison, charring depth development based on the design charring rate of 0.65 mm/min for solid timber is shown.

Based on the test findings, the dovetail structure effectively restrained the delamination of the unloaded DMWBE panel during fire exposure. The calculated average charring rates between the exposed face and 40 mm, as well as between 40 mm and 80 mm, were determined to be 0.65 mm/min and 0.70 mm/min, respectively. These rate values were only marginally higher than those observed in solid timber. In the first layer, the rates of charring exhibited a range of 0.57 mm/min to 0.70 mm/min, while in the secondary layer, they varied between 0.52 mm/min and 0.83 mm/min. Figure 14 depicts the charred dovetail geometry following the test. It should be noted that the charring performance and rates can differ significantly if thinner lamellae are employed, as was the case with the tested structure where the panels consisted of 60 mm thick lamellae. For the CLT board, the charring rate in the first lamella layer was 0.62 mm/min, but as charring progressed into the second lamella layer, the rate escalated to 0.93 mm/min due to char fall-off. This divergence can be observed in Figure 13, where the CLT curve deviates from the curves of solid timber and DMWSE. Within the CLT panel, the rates of charring in the initial layer varied from 0.57 mm/min to 0.69 mm/min, while in the subsequent layer, they ranged from 0.78 mm/min to 1.36 mm/min.

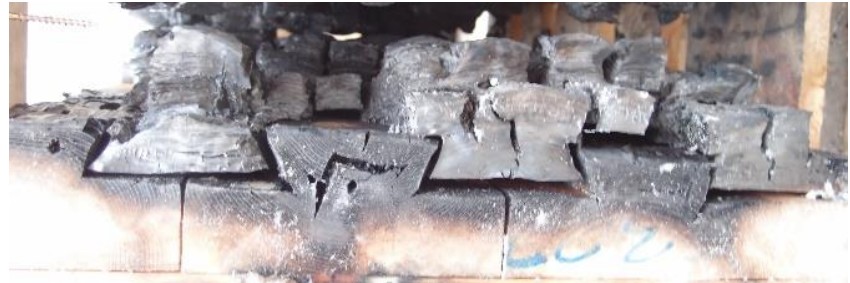

**Figure 14.** Remaining lamella layers and the dovetail structure of DMWBEs at the end of the test.

The research did not investigate uncertainties associated with the thermocouples' position accuracy or the thermal disturbance errors caused by the presence of shielded thermocouples.

## 5. Conclusions

Two fire experiments were carried out on DMWBE and CLT samples with similar lamellae thicknesses to examine the fire/charring performance, thermal insulation properties, and charring behavior of the adhesive-free DMWBE structure. The obtained results

and observations were compared with those of a CLT panel composed of similar lamellae thicknesses produced using a polyurethane adhesive.

Within the existing body of literature, a notable absence pertains to investigations into the fire/charring performance of emerging adhesive-free methodologies like wood dowels, rotary-dowel welding, and wooden nails, as noted in the Introduction. Consequently, it becomes unfeasible to engage in an exhaustive discourse aimed at furnishing insights into the congruences or disparities between the outcomes of our study and those of other research endeavours.

The charring performance of the dovetail construction closely resembled that of solid timber, with only a slightly higher charring rate. Towards the end of the test, the char front was localized in the third of the five lamella layers, while no signs of flames or hot gases were detected on the unexposed side. The dovetail detail effectively prevented char fall-off, as evidenced by the tested lamella thickness. Conversely, the CLT specimens exhibited a noticeable increase in the charring rate, attributed to the fall-off of the first lamellae layer.

It is important to emphasize that our findings is specifically applicable to lamellae with a thickness of 40 mm. Our findings do not provide insights into the performance of panels constructed from thinner lamellae. When DMWBE panels are used as load-bearing structures, the applied external forces cause stresses and deformations, which in turn can affect the performance of the dovetail detail, char fall-off and char depths. Further research is required to investigate the effects of these factors.

Within the scope of the DoMWoB project, an extensive strategy was devised. This strategy encompassed a range of assessments, including evaluations of structural (bending) capabilities, air permeance, sound insulation, and fire performance. The ultimate aim was to advance DMWBEs as a universally feasible substitute for conventional EWPs.

Concerning DMWBEs, it is imperative to direct heightened attention towards the available materials. Additionally, a stringent certification process is imperative to facilitate the introduction of these construction products into the market. Technical guidelines aligning with national design codes are pivotal, serving as guiding principles for manufacturers, designers, and contractors. Hence, it becomes essential to embark on a comprehensive program of research and development that encompasses a wide spectrum of factors. This includes, but is not confined to dimensional stability, vibration response, creep behavior over time, outdoor performance, thermal conductivity, and environmental life-cycle impact assessment.

Furthermore, insights gleaned from the examination of emerging adhesive-free assembly techniques, such as wood welding, wood nails provide inspiration for the potential application of advanced wood-modification technology and machining processes. These innovations hold the promise of revitalizing traditional wood-joining methods.

In summary, the advancement and widespread adoption of DMWBEs within the construction realm necessitate a rigorous approach. This entails intensive research and development, the establishment of a robust certification framework, and strict adherence to technical guidelines. Such a comprehensive strategy is pivotal in realizing DMWBEs' full potential as an innovative and environmentally sustainable construction material.

**Author Contributions:** Conceptualization, H.E.I., M.K., M.A. and M.M.; methodology, H.E.I., M.K., M.A. and M.M.; formal analysis, M.A. and M.M.; investigation, H.E.I., M.K., M.A. and M.M.; data curation, H.E.I., M.K., M.A. and M.M.; writing—original draft preparation, H.E.I., M.K., M.A. and M.M.; writing—review and editing, H.E.I., M.K., M.A. and M.M.; visualization, H.E.I., M.K., M.A. and M.M.; supervision, H.E.I., M.K., M.A. and M.M.; project administration, H.E.I., M.K., M.A. and M.M. All authors have read and agreed to the published version of the manuscript.

**Funding:** This project has received funding (202,680.96 €) from the European Union's Horizon 2020 research and innovation programme under the Marie Skłodowska-Curie grant agreement No [101024593]. This project has also received funding (60,000 + 33,000 €) from the Marjatta and Eino Kolli Foundation for technical performance tests including fire performance, structural bending, air permeance, and sound insulation, and potential patent applications.

**Institutional Review Board Statement:** Not applicable.

**Informed Consent Statement:** Not applicable.

**Data Availability Statement:** Not applicable.

**Conflicts of Interest:** The authors declare no conflict of interest.

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
