# Peer review of "Evaluating Fire Performance: An Experimental Comparison of Dovetail Massive Wooden Board Elements and Cross-Laminated Timber"

_fire, doi:10.3390/fire6090352_

Round 1

Reviewer 1 Report

The research is exciting and addresses the problem of the resistance to combustion of a conventional material obtained by bonding (CLT) and one using dovetail joint technology (DMWBE).

The choice of materials and test methods are correctly selected and described.

The results are interpreted appropriately, and the conclusions drawn follow those obtained.

The only thing missing in the analysis of the results and conclusions presented is a comparison with studies by other authors and other materials.

Reviewer 2 Report

In this paper, an experimental study on the fire resistance of Dovetail Massive Wooden Board Elements and Cross-Laminated Timber is carried out, and the research results have a certain reference value.

The following suggestions are for reference:

1. Generally speaking, the sample size of this experiment is too small, so the author needs to give enough convincing reasons.

2. In Figure 12, the peak value of the grey curve is almost the same, but the black curve (representative value) is very different. What is the reason?

3. The paper's conclusion is too simple, so it is suggested to deepen it.

Reviewer 3 Report

Good job. The subject is very interesting due to the verification of the similarity of the behavior against fire of solid wood dovetail board elements (DMWBEs) with solid wood. When comparing the results with cross laminated timber (CLT) important conclusions can be drawn.

Small adjustments are recommended:

Line 61, 69 and 117 reference.

The ideas in lines 61-67 all correspond to reference 5? If not, cite each idea separately.

On line 91 is specifically talking about this article? Or the DoMWoB project? Characteristics of DMWBE are not investigated for this article. You investigated the behavior in front of the fire….. Make the correction.

The contextualization that is done in the introduction is very clear and complete (make the corrections suggested)

Point 2 called Fire tests, corresponds to the methodology used, it is suggested to change to Materials and Methods, within this point they include point 3 that they called Test set-up.

In summary: 1 Introduction, 2 Materials and methods, 3 Results, 4 Conclusions and discussion

Figure 7 does not correspond to CTL but to DMWBE, change the figure.

In figure 9 mark which is a) and which is b)

In Conclusions, from line 601 to line 605 is not a conclusion.

Deepen the conclusions and add a discussion. Here you can add the future jobs that you mentioned between lines 98 and 102.

Round 2

Reviewer 2 Report

In general, the manuscript was revised, but the question about the number of test pieces was still not answered positively. On the other hand what I am puzzled about in Figure 12 is that the data peaks for the basic test are 80, but why the average curve does not go to 80?
